# Aspect-to-Scope Oriented Multi-view Contrastive Learning for Aspect-based Sentiment Analysis

**Heyan Chai**[1], **Ziyi Yao**[1], **Siyu Tang**[1], **Ye Wang**[2], **Liqiang Nie**[1],
**Binxing Fang**[1], and **Qing Liao**[1,3*]

[1] Harbin Institute of Technology, Shenzhen, China
[2] National University of Defense Technology, China
[3] Peng Cheng Laboratory, Shenzhen, China

{chaiheyan,tangsiyu999,yaoziyi}@stu.hit.edu.cn,
ye.wang@nudt.edu.cn,nieliqiang@hit.edu.cn,fangbx@cae.cn,liaoqing@hit.edu.cn

## Abstract

Aspect-based sentiment analysis (ABSA) aims to align aspects and corresponding sentiment expressions, so as to identify the sentiment polarities of specific aspects. Most existing ABSA methods focus on mining syntactic or semantic information, which still suffers from noisy interference introduced by the attention mechanism and dependency tree when multiple aspects exist in a sentence. To address these issues, in this paper, we revisit ABSA from a novel perspective by proposing a novel scope-assisted multi-view graph contrastive learning framework. It not only mitigates noisy interference for better locating aspect and its corresponding sentiment opinion with aspect-specific scope, but also captures the correlation and difference between sentiment polarities and syntactic/semantic information. Extensive experiments on six benchmark datasets show that our proposed approach substantially outperforms state-of-the-art methods and verifies the effectiveness and robustness of our model.

## 1 Introduction

Aspect-based sentiment analysis (ABSA) is a fine-grained sentiment classification task, aiming at identifying the sentiment polarities (i.e., positive, neutral, or negative) of specific aspects in a given sentence (Wang et al., 2016; Tang et al., 2022; Chen et al., 2022). Different from document- or sentence-level sentiment analysis, different aspects in a sentence may express different sentiment polarities. For example, given a sentence "Service was slow but people were friendly", the sentiment labels of aspect "service" and "people" are negative and positive, respectively. The main challenge of ABSA is how to effectively capture the relationship between aspect term and its corresponding opinion expressions while preventing interference from irrelevant context words.

---
* Corresponding Author

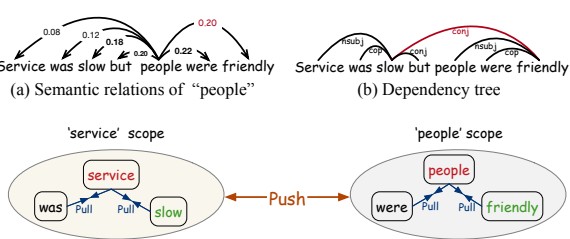

Figure 1: Examples of input sentence "Service was slow but people were friendly". (a) semantic relations of aspect "people" derived by attention mechanism. (b) the dependency tree of input sentence. (c) Our scope-assisted contrastive learning method.

Recent studies on ABSA task either utilize various attention mechanisms to model the semantic relationships between aspect and its context words to obtain aspect-related sentiment representation (Chen et al., 2017; Tan et al., 2019; Oh et al., 2021; Li et al., 2021; Pang et al., 2021), or leverage syntactic information derived from syntax dependency tree to model syntactic structures to capture the structural dependencies between aspect and its sentiment expressions (Sun et al., 2019; Wang et al., 2020; Chen et al., 2020; Zhou et al., 2021).

Despite remarkable progress made by the existing methods, they suffer from the interference of irrelevant context words when modeling the relationship between aspect words and the corresponding sentiment clues. On the one hand, attention-based methods may improperly assign high attention scores to the words irrelevant to the anchor aspect term. As shown in **Figure 1 (a)**, for the anchor aspect "people", its attention score with the corresponding opinion word "friendly" (0.20) is only slightly higher than the word "slow" w.r.t. aspect "service" (0.18), and much lower than that with its neighbor words (e.g. 0.24 for "were"). On the other hand, the dependency tree structure in the syntactic-based methods may also introduce noisy signals, because of the interference among multiple aspects and their corresponding opinions. As shown in **Figure 1 (b)**, it is easy to observe that

the dependency tree builds an unexpected relation between "slow" and "friendly" (see the arc with red color), which are opinion words of different aspects ("service" and "people"). In other words, existing methods are insufficient to accurately capture the relationship between aspect and its corresponding opinion words, discouraging the learning of aspect-dependent sentiment representations.

To address the above issues, we separate different aspect-related context words to prevent the noisy interference from other aspects. Specifically, inspired by the constituent tree-based structure (Xu et al., 2022), we try to outline a structural text region related to a specific target aspect, named *aspect-specific scope*, and ignore all other connections involving the target aspect in the remaining part of the sentence, as shown in **Figure 1 (c)**. The *aspect-specific scope* is expected to cover all the corresponding opinion words and exclude the irrelevant opinion words of other aspects. Based on this idea, we propose a novel aspect-to-scope multi-view contrastive learning (**A2SMvCL**) framework to preferably facilitate the capture of the crucial aspect-dependent sentiment clues to improve the performance of ABSA. More concretely, we first construct two graphs, namely DepG and SemG, to provide syntactic-structure and semantic-correlation from different graph views. Then, a scope-assisted multi-view (intra- and inter-view) graph contrastive learning module is proposed to enhance the model to learn the correlations and differences among aspect-related and aspect-unrelated sentiment representation derived by both views. The scope-based intra-view contrastive learning aims to align each aspect and its corresponding opinion words while separating different aspect-related words (scope) at the same time. The scope-based inter-view contrastive learning aims at further obtaining a richer aspect-dependent sentiment representation by incorporating different types of information from two graph views. Finally, an adaptive multi-graph fusion module is proposed to fuse syntax dependency and semantic information for capturing the syntax- and semantic- enhanced sentiment representation. Our main contributions are summarized as follows:

- The ABSA task is approached from a new perspective that devises the aspect-specific scope to separate different aspect-related context words, so as to preferably eliminate the noisy interference from different aspect-

corresponding opinion words.

- A novel multi-view graph contrastive learning framework is proposed to integrate syntactic dependency and semantic correlation information, facilitating the learning of aspect-dependent sentiment-aware representation.
- Extensive experiments on six public benchmark datasets show the effectiveness and robustness of our proposed A2SMvCL.

## 2  Methodology

In this section, we describe our proposed **A2SMvCL** framework for ABSA in detail. As shown in Figure 2, the proposed A2SMvCL framework mainly contains five components: 1) *Aspect-specific scope Detection*, which detects the scope of the given aspect based on word vectors in a sentence; 2) *Multiple Graphs Generation*, which constructs and encodes two types of graph views for each sentence; 3) *Scope-Assisted Multi-View Graph Contrastive Learning*, which encodes different types of information while filtering out noisy signals from other aspect-specific scopes based on syntactic and semantic graphs. 4) *Adaptive Multi-Graph Fusion*, which fuses syntactic and semantic information, and outputs the final representations; 5) *Aspect-oriented Sentiment Classification*, which classifies the sentiments of given aspect.

### 2.1  Task Definition

Given a sentence $s$ consists of $n$ words $s = [w_1, ..., w_{t+1}, ..., w_{t+m}, ..., w_n]$ and its corresponding aspect $a = [w_{t+1}, ..., w_{t+m}]$, aspect-based sentiment classification aims to identify the sentiment polarity (i.e. *Positive*, *Negative*, or *Neutral*) of this aspect, where $w_i$ is the $i$-th word. Moreover, for each aspect in a sentence, we design an auxiliary aspect-specific scope detection task to identify a sequence of aspect-related terms $Y^a = [y_1^a, y_2^a, ..., y_n^a]$, where $y_i^a \in \{B, I, O\}$ annotated by Xu et al. (2022) denotes the beginning of, inside of, and outside of aspect-related words.

### 2.2  Text Encoder

We adopt a pre-trained language model BERT (Devlin et al., 2019) as encoder to encode both sentence $s$ and its corresponding aspect $a$. We follow previous works to construct a BERT-based sentence-aspect pair sequence "$[CLS]s[SEP]a[SEP]$" and feed the sequence into BERT to acquire the contextualized representation $\boldsymbol{H} =$

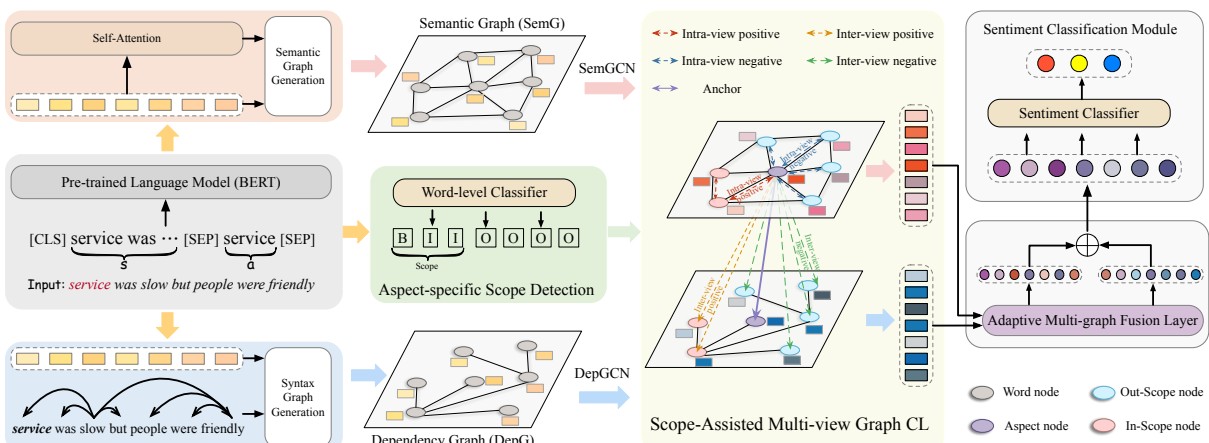

Figure 2: The architecture of the proposed A2SMvCL framework.

$[\boldsymbol{h}_1, ..., \boldsymbol{h}_n] \in \mathbb{R}^{n \times d_m}$, defined by $\boldsymbol{H} = \text{BERT}([CLS]s[SEP]a[SEP])$, where $\boldsymbol{h}_i \in \mathbb{R}^{d_m}$ is the hidden representation of the $i$-th word.

## 2.3 Aspect-specific Scope Detection

To accurately identify the aspect-related context words, we introduce a constituent tree-based structure, *aspect-specific scope*, to separate aspect-related and aspect-unrelated words. *Aspect-specific scope* is the smallest continuous sub-region of the input sentence, containing an aspect and the corresponding sentiment expression. Specifically, given word vector matrix $\boldsymbol{H}$ learned by the *Text Encoder*, we utilize a linear-chain conditional random fields (CRF) layer (Lafferty et al., 2001) to predict the aspect-specific scope, which can be formulated as:

$$Y^a = \text{CRF}(\boldsymbol{H}) \quad (1)$$

where $Y^a = [y_1^a, .., y_n^a]$, and $y_i^a \in \{B, I, O\}$. Note that, $y_i^a = B$ or $I$ denotes the $i$-th word of sentence $s$ is in aspect-specific scope, and $y_i^a = O$ indicates the $i$-th word is not in aspect-specific scope. The objective of aspect-specific scope detection is minimizing the negative log-likelihood (NLL) loss:

$$\mathcal{L}_{scope} = -\sum \log p(Y^a | \boldsymbol{H}) \quad (2)$$

## 2.4 Multiple Graph Generation

To leverage the syntactic structure and semantic correlation information of the input sentence, we design *syntax graph construction* and *semantic graph construction* modules. We feed the generated Dependency Graph (DepG) containing syntactic structure information obtained by dependency tree and Semantic Graph (SemG) containing rich semantic correlation information obtained by self-attention mechanism into two GCN encoders for acquiring aspect-dependent graph representations.

**Syntax-based Graph Construction.** For an input sentence $s$, we obtain the corresponding dependency tree $\mathcal{T}^s$ over $s$ through syntactical dependency parser[1]. We subsequently build an undirected graph $\boldsymbol{G}^{dep}$ (dubbed DepG) by taking each word in $s$ as a node and representing head-dependent relations in $\mathcal{T}^s$ as edges. Formally, the adjacent matrix $\boldsymbol{A}^{dep} \in \mathbb{R}^{n \times n}$ is defined by

$$\boldsymbol{A}_{ij}^{dep} = \begin{cases} 1 & \text{if } \text{link}(i,j) = 1 \text{ or } i = j, \\ 0 & \text{otherwise.} \end{cases} \quad (3)$$

where $\text{link(i, j)}$ means $i$-th and $j$-th words have dependency relation. To encode the syntactic information of $\boldsymbol{G}^{dep}$, we employ a GCN module (named *DepGCN*) to capture richer syntactic graph representation $\boldsymbol{H}^{dep}$,

$$\boldsymbol{H}_l^{dep} = \text{ReLU}(\boldsymbol{A}^{dep}\boldsymbol{H}_{l-1}^{dep}\boldsymbol{W}_l^{dep} + \boldsymbol{b}_l^{dep}) \quad (4)$$

where $\boldsymbol{W}_l^{dep} \in \mathbb{R}^{d_m \times d_m}$ and $\boldsymbol{b}_l^{dep} \in \mathbb{R}^{d_m}$ are model parameters for the $l$-th layer. The initial input $\boldsymbol{H}_0^{dep}$ of first GCN is derived by the node representations $\boldsymbol{H}$ learned by *Text Encoder* in Section 2.2, i.e. $\boldsymbol{H}_0^{dep} = \boldsymbol{H} = [\boldsymbol{h}_1, ..., \boldsymbol{h}_n]$.

**Semantic-based Graph Construction.** Additionally, to capture the semantic correlations among words in a sentence, we build a directed graph $\boldsymbol{G}^{sem}$ (dubbed SemG) by taking each word as a node and the similarity scores between two arbitrary words computed by self-attention mechanism as edges. Formally, the adjacent matrix $\boldsymbol{A}^{sem} \in \mathbb{R}^{n \times n}$ is given by

$$\boldsymbol{A}^{sem} = \text{softmax}\left(\frac{(\boldsymbol{Q}\boldsymbol{W}_q)(\boldsymbol{K}\boldsymbol{W}_k)^T}{\sqrt{d_m}}\right) \quad (5)$$

where $\boldsymbol{Q}$ and $\boldsymbol{K}$ are two copies of $\boldsymbol{H}$, representing query and key vectors, respectively. $\boldsymbol{W}_k, \boldsymbol{W}_q \in$

---

[1]We use spaCy toolkit: https://spacy.io/

$\mathbb{R}^{d_m \times d_m}$ are model parameters. $d_m$ is the dimensionality of node feature. Similar to DepG, we also employ a SemGCN module (same structure with DepGCN but does not share parameters) to capture richer semantic representation $\boldsymbol{H}^{sem}$ of SemG.

## 2.5 Scope-Assisted Multi-View Graph Contrastive Learning

Based on the generated DepG and SemG learned over syntax dependency tree and semantic correlation information, here we discuss how to effectively leverage these two types of information to capture richer aspect-dependent sentiment representation. To mitigate the noise like irrelevant relations and irrational attention scores brought up by DepG and SemG, we introduce *aspect-specific scope*, and design a novel scope-assisted multi-view graph contrastive learning to align each aspect and its corresponding opinion words while separating in- and out-scope words at the same time. Specifically, intra-view contrastive learning is conducted within each graph, and inter-view contrastive learning is processed between two graphs.

### 2.5.1 Intra-View Contrastive Learning

To better align each aspect and its corresponding opinion words while separating different aspect-related words (scope) at the same time, we define an intra-view contrastive loss on the node representations generated by DepG and SemG views based on aspect-specific scope. For ease of distinction, we denote node representations[2] in DepG and SemG views as $\{\boldsymbol{u}_i\}_{i \in [n]}$ and $\{\boldsymbol{v}_i\}_{i \in [n]}$, respectively. We denote aspect-specific scope in DepG and SemG views as $\mathcal{S}^{\boldsymbol{u}}$ and $\mathcal{S}^{\boldsymbol{v}}$, respectively. Taking the SemG view in Figure 2 as an example, we define the aspect node representation $\boldsymbol{v}_i$ as the *anchor*. For any node $v_j$ in aspect-specific scope $\mathcal{S}^{\boldsymbol{v}}$ ($v_j \in \mathcal{S}^{\boldsymbol{v}}$), its representation $\boldsymbol{v}_j$ forms the *positive* sample, and any node $v_k$ not in $\mathcal{S}^{\boldsymbol{v}}$ ($v_k \notin \mathcal{S}^{\boldsymbol{v}}$) is naturally regarded as *negative* sample. Then the contrastive loss is computed across all *positive* paris, $(\boldsymbol{v}_i, \boldsymbol{v}_j)$ in a sentence by mirroring the InfoNCE (van den Oord et al., 2018) objective,

$$\mathcal{L}^V_{intra}(\boldsymbol{v}_i) = -\log \frac{\sum_{j \in \mathcal{S}^{\boldsymbol{v}} \setminus i} e^{\varphi(\boldsymbol{v}_i, \boldsymbol{v}_j)/\tau}}{\sum_{k \neq i} e^{\varphi(\boldsymbol{v}_i, \boldsymbol{v}_k)/\tau}} \quad (6)$$

where $\tau$ is a temperature parameter. We define the critic $\varphi(\cdot, \cdot) = \text{sim}(\sigma(\cdot), \sigma(\cdot))$, where $\text{sim}(\cdot, \cdot)$ is

---

[2]Note that, $\boldsymbol{u}_i$ and $\boldsymbol{v}_i$ are the $i$-th vector of $\boldsymbol{H}^{dep}$ and $\boldsymbol{H}^{sem}$ respectively, defined in Section 2.4.

the cosine similarity and $\sigma(\cdot)$ is a nonlinear projection to enhance the expression power of the critic.

Similarly, we calculate $\mathcal{L}^U_{intra}$ for the DepG view. Finally, by combining the two losses, we have the intra-view contrastive learning objective function in each mini-batch $\mathcal{B}$, which can be defined as:

$$\mathcal{L}_{intra} = \frac{1}{N_{\mathcal{B}}} \sum_{\boldsymbol{v}_i, \boldsymbol{u}_i \in \mathcal{B}} (\mathcal{L}^V_{intra}(\boldsymbol{v}_i) + \mathcal{L}^U_{intra}(\boldsymbol{u}_i)) \quad (7)$$

### 2.5.2 Inter-View Contrastive Learning

To further enhance the sentiment representation of aspect-related embedding, we perform contrastive learning between two graph views. This takes the correlation among different types of graph views into consideration while preferentially performing the contrastive learning learning of aspect-related word representation.

More concretely, taking the SemG view as an example, we define the aspect node representation $\boldsymbol{v}_i$ as the *anchor*, its corresponding representation $\boldsymbol{u}_i$ and aspect-specific scope $\mathcal{S}^{\boldsymbol{u}}$ generated in another view are treated as *positive* samples. Naturally, we define all remaining nodes in another view as *negative* samples. Moreover, to integrate the importance between different positive sample pairs into contrastiveness, we use the similarity $\omega$ of two views to re-weight positive sample pairs to improve the diversity of positive samples. The contrastive loss for SemG view is computed as follow:

$$\ell(\boldsymbol{v}_i, \boldsymbol{u}_i) = -\log \Big( \frac{e^{\varphi(\boldsymbol{v}_i, \boldsymbol{u}_i)/\tau} + \sum_{j \in \mathcal{S}^{\boldsymbol{u}} \setminus i} e^{\omega \cdot \varphi(\boldsymbol{v}_i, \boldsymbol{u}_j)/\tau}}{\sum_{k=1}^n e^{\varphi(\boldsymbol{v}_i, \boldsymbol{u}_k)/\tau}} \Big)$$
$$(8)$$

We define the similarity of the two views $\omega = \text{sim}(\boldsymbol{v}_i, \boldsymbol{u}_i)$, where $\boldsymbol{v}_i$ and $\boldsymbol{u}_i$ are aspect node representations learned by SemG and DepG, respectively.

Since the two views are symmetric, the loss of the DepG view $\ell(\boldsymbol{u}_i, \boldsymbol{v}_i)$ is similarly defined as $\ell(\boldsymbol{v}_i, \boldsymbol{u}_i)$. The overall inter-view contrastive objective in each mini-batch $\mathcal{B}$ is formally given as:

$$\mathcal{L}_{inter} = \frac{1}{2N_{\mathcal{B}}} \sum_{\boldsymbol{v}_i, \boldsymbol{u}_i \in \mathcal{B}} (\ell(\boldsymbol{v}_i, \boldsymbol{u}_i) + \ell(\boldsymbol{u}_i, \boldsymbol{v}_i)) \quad (9)$$

By combining the intra- and inter-view contrastive learning, the overall loss of scope-assisted multi-view graph contrastive learning module is :

$$\mathcal{L}_{gcl} = \mathcal{L}_{intra} + \mathcal{L}_{inter}. \quad (10)$$

## 2.6 Adaptive Multi-Graph Fusion

Based on the graph representation of DepG and SemG, $\boldsymbol{H}^{dep}$ and $\boldsymbol{H}^{sem}$, derived by dependency

tree and semantic correlation information, here we propose an adaptive multi-graph fusion module to adaptively fuse syntax dependency and semantic information for capturing syntax- and semantic-enhanced aspect-dependent output representation,

$$\widehat{\boldsymbol{H}}^{dep} = \text{ReLU}(\boldsymbol{W}(\alpha \boldsymbol{H}^{dep} + \beta \boldsymbol{H}^{sem})) \quad (11)$$

$$\alpha = \rho \cdot \sigma(g(\boldsymbol{H}^{dep})), \ \ \beta = 1 - \alpha \quad (12)$$

where $\alpha$ and $\beta$ are the dynamic fusing proportions, $g(\cdot)$ is a self-gating function with a shared convolutional kernel. $\rho \in [0, 1]$ is the hyper-parameter of prior knowledge.

Since the two graph are symmetric, the fused SemG representation $\widehat{\boldsymbol{H}}^{sem}$ is defined similarly as $\widehat{\boldsymbol{H}}^{dep}$. Finally, we concatenate DepG representation $\widehat{\boldsymbol{H}}^{dep}$ and SemG representation $\widehat{\boldsymbol{H}}^{sem}$ to obtain syntax- and semantic-enhanced aspect-dependent representation $\boldsymbol{r}$:

$$\boldsymbol{r} = \widehat{\boldsymbol{H}}^{dep} \oplus \widehat{\boldsymbol{H}}^{sem}. \quad (13)$$

where $\boldsymbol{r} \in \mathbb{R}^{n \times 2d_m}$.

## 2.7 Model Training

**Aspect-oriented Sentiment Classification.** By applying average pooling on aspect node representations of $\boldsymbol{r}$, we get the final aspect-oriented representation vector $\boldsymbol{z}$:

$$\boldsymbol{z} = \text{AvgPooling}(\boldsymbol{r}_{a_1}, \boldsymbol{r}_{a_2}, \cdots, \boldsymbol{r}_{a_m}) \quad (14)$$

where $\boldsymbol{r}_{a_i}$ denotes the $i$-th word in aspect $a$ of sentence $s$. After that, we adopt a fully-connected layer with softmax normalization to yield a probability distribution of sentiment representation:

$$\widehat{\boldsymbol{y}} = \text{softmax}(\boldsymbol{W}_s \boldsymbol{z} + \boldsymbol{b}_s) \quad (15)$$

where $\widehat{\boldsymbol{y}} \in \mathbb{R}^{d_p}$ is the predicted sentiment polarity, $d_p$ is the dimensionality of sentiment labels. $\boldsymbol{W}_s \in \mathbb{R}^{d_p \times d_m}$ and $\boldsymbol{b}_s \in \mathbb{R}^{d_p}$ are parameters to be learned. Then the cross-entropy loss between the predicted labels and the ground-truth label $\boldsymbol{y}$ of $i$-th instance is defined as:

$$\mathcal{L}_{cls} = -\sum_{i=1}^{N_{\mathcal{B}}} \sum_{j=1}^{d_p} \boldsymbol{y}_i^j \log \widehat{\boldsymbol{y}}_i^j \quad (16)$$

**Joint Learning Objective.** The learning objective of our proposed model is to train model by jointly minimizing the three losses generated by sentiment classification, aspect-specific scope detection, and multi-view graph contrastive learning, which can be formulated as:

$$\mathcal{L}(\Theta) = \gamma_c \mathcal{L}_{cls} + \gamma_s \mathcal{L}_{scope} + \gamma_g \mathcal{L}_{gcl} + \lambda ||\Theta||^2 \quad (17)$$

| Dataset | #Positve | | #Negative | | #Neutral | |
|---|---|---|---|---|---|---|
| | Train | Test | Train | Test | Train | Test |
| LAP14 | 976 | 337 | 851 | 128 | 455 | 167 |
| REST14 | 2164 | 727 | 807 | 196 | 637 | 196 |
| REST15 | 912 | 326 | 36 | 34 | 256 | 182 |
| REST16 | 1657 | 611 | 101 | 44 | 748 | 204 |
| TWITTER | 1507 | 172 | 1528 | 169 | 3016 | 336 |
| MAMS | 3380 | 400 | 2764 | 329 | 5042 | 607 |

Table 1: Satistics on six datasets of ABSA.

where $\gamma_c$, $\gamma_s$, and $\gamma_g$ are the coefficients. $\Theta$ represents all trainable parameters. $\lambda$ represents the coefficient of $L_2$-regularization.

## 3 Experimental Setup

### 3.1 Datasets

We conduct experiments on six public standard datasets and the detailed statistics are shown in **Table 1**. The REST14, LAP14, REST15 and REST16 datasets are from SemEval ABSA challenges (Pontiki et al., 2014, 2015, 2016), consisting of reviews in the restaurant and laptop domains. The TWITTER dataset is a collection of tweets (Dong et al., 2014). The MAMS dataset is consisted of sentences with multiple aspects (Jiang et al., 2019). Furthermore, in order to use *scope* labels, we use the expanded versions of these six datasets from Xu et al. (2022).

### 3.2 Implementation Details

We use Stanford parser[3] to get syntactic dependency trees and employ the pre-trained BERT[4] in PyTorch. To alleviate overfitting, we set dropout at a rate of 0.3, learning rate at $2 \times 10^{-5}$ of adam optimizer and a $L_2$ regulation at $10^{-5}$ for training. The layers of DepGCN and SemGCN are 2. The parameters $\gamma_c$ and $\gamma_s$ are set to 1 and 0.03 for all datasets. The hyper-parameters $\gamma_g$ and $\tau$ are respectively set to (0.4, 0.11), ( 0.22, 0.13), (0.81, 0.1), (0.2, 0.14) and (0.19, 0.12) for the five datasets. Accuracy(Acc.) and macro-f1 (F1.) are evaluation metrics.

### 3.3 Baselines

We compare the proposed **A2SMvCL** with a series of strong ABSA baselines, including semantic

---

[3] https://stanfordnlp.github.io/CoreNLP/
[4] https://github.com/huggingface/transformers

| Model | Lap14 | | Rest14 | | Rest15 | | Rest16 | | Twitter | | MAMS | |
|---|---|---|---|---|---|---|---|---|---|---|---|---|
| | Acc.(%) | F1.(%) | Acc.(%) | F1.(%) | Acc.(%) | F1.(%) | Acc.(%) | F1.(%) | Acc.(%) | F1.(%) | Acc.(%) | F1.(%) |
| BERT-SRC | 80.56 | 77.20 | 84.55 | 75.74 | 83.03 | 63.92 | 90.75 | 74.00 | 73.41 | 72.38 | 82.82 | 81.90 |
| SDGCN | 81.35 | 78.34 | 83.57 | 76.47 | - | - | - | - | - | - | - | - |
| BATAE-GRU | 78.59 | 74.78 | 84.11 | 76.09 | - | - | - | - | 74.34 | 72.76 | - | - |
| IMA | 77.44 | 73.48 | 82.81 | 73.66 | 79.29 | 64.41 | 83.24 | 64.63 | - | - | - | - |
| R-GAT | 78.21 | 74.07 | 86.60 | 80.16 | - | - | - | - | 76.15 | 74.88 | 84.52 | 83.74 |
| KumaGCN | 81.98 | 78.81 | 86.43 | 80.30 | 86.35 | 70.76 | 92.53 | 79.24 | 77.89 | 77.03 | - | - |
| ACLT | 79.68 | 75.83 | 85.71 | 78.44 | 84.44 | 72.08 | 92.15 | 78.64 | 75.48 | 74.51 | - | - |
| HGCN | 79.59 | - | 86.45 | - | 83.91 | - | 91.72 | - | - | - | - | - |
| dotGCN | 81.03 | 78.10 | 86.16 | 80.49 | 85.24 | 72.74 | 93.18 | 82.32 | 78.11 | 77.00 | 84.95 | 84.44 |
| BiSyn-GAT | 79.43 | 75.07 | 86.70 | 79.57 | 83.39 | 71.72 | 88.82 | 71.75 | 76.51 | 75.34 | - | - |
| DGEDT | 79.80 | 75.60 | 86.30 | 80.00 | 84.00 | 71.00 | 91.90 | 79.00 | 77.90 | 75.40 | - | - |
| BERT4GCN | 77.49 | 73.01 | 84.75 | 77.11 | 83.23 | 67.27 | 87.78 | 75.34 | 74.73 | 73.76 | - | - |
| DualGCN | 81.80 | 78.10 | 87.13 | 81.16 | 84.69 | 72.97 | 89.87 | 77.26 | 77.40 | 76.02 | - | - |
| CPA-SA | 75.18 | 71.5 | 82.64 | 73.38 | - | - | - | - | - | - | - | - |
| MGFN | 81.83 | 78.26 | 87.31 | 82.37 | 84.40 | 72.66 | 92.04 | 81.57 | 78.29 | **77.27** | - | - |
| **A2SMvCL (Ours)** | **82.12** | **78.82** | **87.86** | **82.41** | **86.74** | **75.05** | **93.42** | **83.80** | **78.49** | 77.18 | **85.10** | **84.65** |

Table 2: Experimental results comparison on five publicly available ABSA datasets. The results with "-" denote that no results were reported or code was not released in the original paper.

information based methods: **BERT-SPC** (Devlin et al., 2019), **SDGCN** (Zhao et al., 2020), **BATAE-GRU** (Wang and Wang, 2021), and **IMA** (Wang et al., 2022), syntax information based methods: **R-GAT** (Wang et al., 2020), **KumaGCN** (Chen et al., 2020), **ACLT** (Zhou et al., 2021), **HGCN** (Xu et al., 2022), **dotGCN** (Chen et al., 2022), and **BiSyn-GAT** (Liang et al., 2022), both semantic and syntax information based methods: **DGEDT** (Tang et al., 2020), **BERT4GCN** (Xiao et al., 2021), **DualGCN** (Li et al., 2021), **CPA-SA** (Huang et al., 2022), and **MGFN** (Tang et al., 2022).

In addition, we also design several variants of our proposed **A2SMvCL** in the ablation study. "***w/o* $\mathcal{L}_{gcl}$**" denotes without all contrastive learning. "***w/o* $\mathcal{L}_{intra}$**" and "***w/o* $\mathcal{L}_{inter}$**" denote without intra- and inter-view contrastive learning, respectively. "***w/o* Adaptive Fusion**" denotes without *Adaptive Multi-Graph Fusion* module. "***w/o* $Scope$**" denotes without considering scope, simply performing graph contrastive learning on all the node representations learned by SemG and DepG.

## 4 Experimental Results

### 4.1 Main Results

The main comparison results of ABSA on five datasets are reported in **Table 2**. It can be observed from the experimental results that our proposed A2SMvCL framework consistently outperforms all comparison baselines on the Lap14, Rest14, Rest15, Rest16 datasets, and MAMS datasets, and achieves an overall better performance than comparison baselines on the Twit-

ter dataset. These indicate the effectiveness of our proposed A2SMvCL in the ABSA task. More concretely, compared with semantic information based methods such as BERT-SPC and SDGCN, our A2SMvCL framework removes the noise interference from the irrelevant opinion words that could be introduced through the attention mechanism. Moreover, compared with the syntax information based methods, such as ACLT and dotGCN, our A2SMvCL improves significantly across all datasets on both metrics. This verifies that exploring scope-assisted graph contrastive learning to separate different aspect-related opinion words can better facilitate the capturing of aspect-dependent sentiment clues, so as to improve the performance of ABSA. In addition, compared with both semantic and syntax information based methods, our model achieves better performance. This demonstrates that our model can better capture aspect-oriented semantic and syntax information as well as eliminate the interference introduced by the attention mechanism and dependency tree structure.

### 4.2 Ablation Study

To further investigate the role of different components in our **A2SMvCL** model, we conduct extensive ablation studies and report the results in **Table 3**. We observe that removing the scope-assisted multi-view graph contrastive learning ("***w/o* $\mathcal{L}_{gcl}$**", "***w/o* $\mathcal{L}_{intra}$**", and "***w/o* $\mathcal{L}_{inter}$**") sharply reduces the performance across all datasets. This demonstrates that intra- and inter-view contrastive learning based on aspect-specific scopes can help the model ignore

| Model | LAP14 | | REST14 | | REST15 | | REST16 | | TWITTER | |
|---|---|---|---|---|---|---|---|---|---|---|
| | Acc.(%) | F1.(%) | Acc.(%) | F1.(%) | Acc.(%) | F1.(%) | Acc.(%) | F1.(%) | Acc.(%) | F1.(%) |
| **A2SMvCL (ours)** | **82.12** | **78.82** | **87.86** | **82.41** | **86.74** | **75.05** | **93.42** | **83.80** | **78.49** | **77.18** |
| *w/o* $\mathcal{L}_{gcl}$ | 79.65 | 76.48 | 86.05 | 80.38 | 84.86 | 71.83 | 92.11 | 79.37 | 77.18 | 75.58 |
| *w/o* $\mathcal{L}_{intra}$ | 79.97 | 76.72 | 86.96 | 80.84 | 85.79 | 72.03 | 93.09 | 83.10 | 77.88 | 76.65 |
| *w/o* $\mathcal{L}_{inter}$ | 80.29 | 77.32 | 86.16 | 80.49 | 86.55 | 72.64 | 92.43 | 82.78 | 77.18 | 76.18 |
| *w/o* $Scope$ | 80.57 | 77.40 | 86.70 | 81.42 | 85.80 | 74.77 | 92.27 | 80.57 | 78.20 | 76.86 |
| *w/o* $Adaptive\ Fusion$ | 81.41 | 77.60 | 87.14 | 81.57 | 86.36 | 74.64 | 92.76 | 82.45 | 77.47 | 76.00 |

Table 3: Experimental results of ablation study.

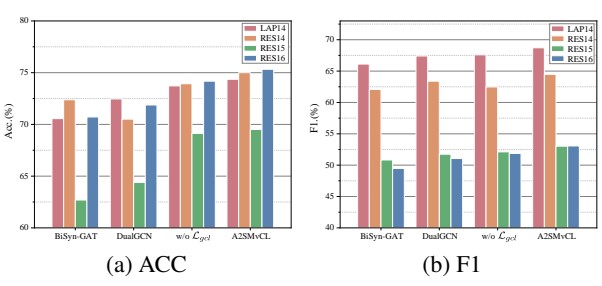

(a) ACC  (b) F1

Figure 3: Results of model that training on the TWIT-TER dataset, and testing the trained model on the other four datasets.

the noise interference from other aspect-specific scopes, improving the quality of aspect-dependent sentiment representation which in turn results in a better performance of the ABSA task. It can also be observed that the performance degrades significantly on all datasets when removing the scope (***w/o*** $Scope$) and performing vanilla graph contrastive learning on all node representations of two graph views. This indicates that scope can effectively separate aspect-based context words. Additionally, A2SMvCL without adaptive multi-graph fusion module ("***w/o*** $Adaptive\ Fusion$") performs slightly worse, which implies that incorporating DepG and SemG representations together can further improve the performance of ABSA.

### 4.3 Robustness Study

We further conduct experiments in the cross-domain scenario to analyze the robustness of our A2SMvCL. To be more specific, we first train a source model on the TWITTER dataset and test its performance on the other four datasets. As shown in **Figure 3**, our model outperforms three comparison methods across all four datasets, illustrating the strength of our proposed A2SMvCL in learning the aspect-dependent sentiment representation. This implies that the scope-assisted multi-view graph contrastive learning framework has strong robustness for grasping aspect-opinion interactions.

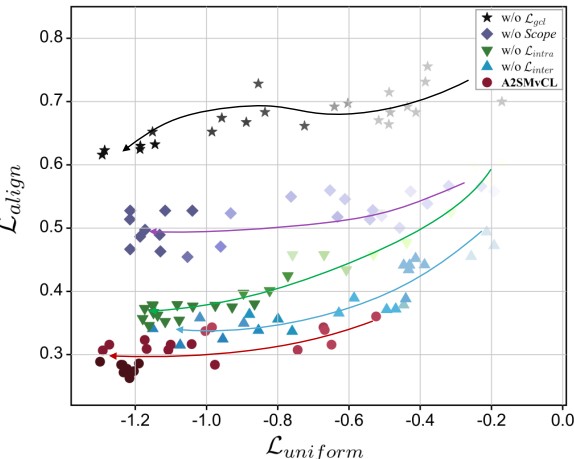

Figure 4: Visualization of contrastive representation. The darker the color of the point, the greater the accuracy. The arrows indicate the training direction. Bottom-left is better.

### 4.4 Analysis

**Analysis of Contrastive Representation.** To further analyze how the proposed A2SMvCL works in contrastive representation learning, we train the four variants of A2SMvCL and A2SMvCL itself and visualize the alignment and uniformity metrics in **Figure 4**. We follow Wang and Isola (2020) and use the same evaluation metrics, *alignment* $\mathcal{L}_{align}$ and *uniformity* $\mathcal{L}_{uniform}$, where lower $\mathcal{L}_{align}$ and $\mathcal{L}_{uniform}$ denotes better performance of the model. Compared with the four variants of A2SMvCL, our model has the lowest $\mathcal{L}_{align}$ and $\mathcal{L}_{uniform}$ during training, which indicates that our A2SMvCL attains strong ability in contrastive learning. Compared with "***w/o*** $\mathcal{L}_{intra}$" and "***w/o*** $\mathcal{L}_{inter}$"), A2SMvCL ***w/o*** $Scope$ presents worse alignment and uniformity, indicating that scope is helpful to obtain a better aspect-dependent sentiment representation. This implies the effectiveness and superiority of our proposed scope-assisted multi-view graph contrastive learning.

**Case Study.** To investigate how A2SMvCL works in eliminating noisy interference from other

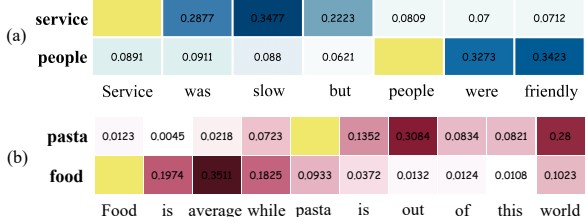

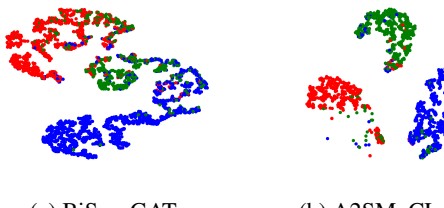

| | | | | | | | |
|---|---|---|---|---|---|---|---|
| (a) service | | 0.2877 | 0.3477 | 0.2223 | 0.0809 | 0.07 | 0.0712 |
| people | 0.0891 | 0.0911 | 0.088 | 0.0621 | | 0.3273 | 0.3423 |
| | Service | was | slow | but | people | were | friendly |

| | | | | | | | | | | |
|---|---|---|---|---|---|---|---|---|---|---|
| (b) pasta | 0.0123 | 0.0045 | 0.0218 | 0.0723 | | 0.1352 | 0.3084 | 0.0834 | 0.0821 | 0.28 |
| food | | 0.1974 | 0.3511 | 0.1825 | 0.0933 | 0.0372 | 0.0132 | 0.0124 | 0.0108 | 0.1023 |
| | Food | is | average | while | pasta | is | out | of | this | world |

Figure 5: Case study of samples with multiple aspects. (a) Both "service" and "people" are aspects. (b) "pasta" and "food" are two aspect terms.

|  (a) BiSyn-GAT  |  (b) A2SMvCL (Ours)  |
|---|---|

Figure 6: Visualization of intermediate embeddings on the LAP14 dataset. Blue dots denote *Positive* examples, red dots denote *Negative* examples, and green dots denote *Neutral* examples. More results in Appendix B.

aspect-related opinions, we conduct a case study and visualize the attention weights of words for each aspect in **Figure 5**. We can observe that the weights between aspect and its corresponding opinion words are higher than ones between aspect and aspect-unrelated words (like "service" and "slow", "service" and "friendly"). This implies that our proposed A2SMvCL can indeed separate the opinion words from distinct aspect-related words and accurately grasp the aspect-opinion interactions, so as to improve the performance of ABSA. Due to the page limitation, we place more case studies in Appendix A.

**Visualization.** To more intuitively verify the effectiveness of our model, we utilize t-SNE (Van der Maaten and Hinton, 2008) to visualize the intermediate embeddings of aspect-dependent sentiment representations learned by BiSyn-GAT and our proposed A2SMvCL on LAP14 dataset. As shown in **Figure 6**, it can be seen that the embeddings within the same sentiment are more aggregated and the boundaries between different sentiments are more distinct for A2SMvCL. This verifies that A2SMvCL can derive more definite correlations and clearer differences among different sentiments, which facilitates the ABSA task. See more visualization samples in Appendix B.

## 5 Related Works

Early efforts in ABSA mainly use the traditional machine learning techniques, such as bag-of-words, SVM, and so on, to manually extract engineered features to classify the sentiment (Wang et al., 2011; Hegde and Seema, 2017). Recently, some methods use attention mechanism to explore the word-level semantic correlations between contextual words (Chen et al., 2017; Fan et al., 2018; Gu et al., 2018; Devlin et al., 2019; Zhao et al., 2020; Wang and Wang, 2021; Wang et al., 2022; Xiao et al., 2021), which aims at capturing the word-level semantic interaction between aspect and context. Some methods use syntax dependency trees to model the syntactic relationship between aspect and its corresponding opinion. For example, Zhang et al. (2019), Wang et al. (2020) and Liang et al. (2022) utilize aspect-oriented dependency trees to capture the syntactic relationship with graph neural network. Chen et al. (2020) uses GCN to incorporate syntactic information and HardKuma-based latent graph information. Xu et al. (2022) introduces the concept of scope, which outlines a structural text region related to a specific aspect.

Moreover, some works (Tang et al., 2020; Li et al., 2021; Tang et al., 2022) utilize GCN to effectively fuse syntactic and semantic information to obtain richer aspect-oriented sentiment representation. Our work shares a similar syntactic and semantic information extraction approach, however we focus on designing the multi-view graph contrastive learning to capture the significant aspect-dependent sentiment clues to improve ABSA task.

## 6 Conclusion

We have re-examined the challenges encountered by existing models for ABSA, and pointed out the suitability of the scope-assisted graph contrastive learning for addressing these issues. Accordingly, we have proposed a novel aspect-to-scope oriented multi-view contrastive learning framework to mitigate noisy interference, and leverage the correlations and differences among different sentiment polarities and different types of information (syntactic dependency and semantic correlation) Extensive experiments on the six benchmark datasets have demonstrated the effectiveness and superiority of our proposed model.

## Limitations

The major limitation is that our method is not suitable for simultaneously identifying multiple sentiment polarities of multiple aspects in one sentence. Actually, this is a common limitation of existing ABSA methods. In our A2SMvCL framework, we generate semantic and syntax graphs. Both two graphs can provide richer information to advance a better performance of ABSA task. For an input sentence with very long text, this could result in increasing of the number of generated graph nodes. Accordingly, the training time of our model will increase, which it would prevent our A2SMvCL from applying to the scenarios with document data.

## Acknowledgements

This work was supported by the National Natural Science Foundation of China (No. 62076079), and The Major Key Project of PCL(Grant No. PCL2022A03), and Guangdong Provincial Key Laboratory of Novel Security Intelligence Technologies (2022B1212010005), and Shenzhen Science and Technology Program (Grant No. ZDSYS20210623091809029).

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

| # | Sentence | DualGCN | BiSyn-GAT | A2SMvCL |
|---|----------|---------|-----------|---------|
| 1 | Other than the crappy service from two individuals, it 's great. | (P✗) | (N✓) | (N✓) |
| 2 | Even fancy ingredients don't make for good pizza unless someone knows how to get the crust right. | (P✗) | (P✗) | (N✓) |
| 3 | Service is highly refined but our seating was delayed 35 minutes past our reservation. | (P✓, O✗) | (P✓, N✓) | (P✓, N✓) |
| 4 | Usually the waiters are kind enough to split the dish in half. | (P✓, P✗) | (P✓, P✗) | (P✓, O✓) |
| 5 | The food is all-around good, with the rolls usually excellent and the sushi not quite on the same level. | (P✓, P✓, P✗) | (P✓, P✓, P✗) | (P✓, P✓, N✓) |
| 6 | Not only is the service great, but atmosphere can easy to form conversation around a table. | (P✓, N✗) | (P✓, N✗) | (P✓, P✓) |

Table 4: Case studies of our A2SMvCL model compared with other baselines.

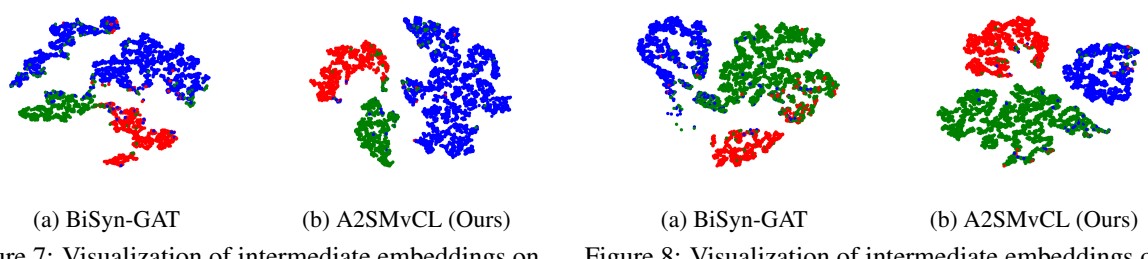

(a) BiSyn-GAT     (b) A2SMvCL (Ours)

Figure 7: Visualization of intermediate embeddings on the REST14 dataset.

(a) BiSyn-GAT     (b) A2SMvCL (Ours)

Figure 8: Visualization of intermediate embeddings on the TWITTER dataset.

## A  More Case Studies

**Table 4** shows the aspect-based sentiment predictions of some cases by using different comparison models, where the symbols *P*, *N*, and *O* represent positive, negative, and neutral sentiments respectively. The words with red color and blue color represent the aspect and its corresponding opinion expressions respectively. Note that, the samples with id from *#3* to *#6* contain multiple aspects, in which the fifth sentence contains three aspects. For the single aspect case (like second sentence *#2*), only our proposed A2SMvCL model predicts it as negative. For DualGCN that using attention mechanism and BiSyn-GAT that using dependency tree, they fail to capture the important sentiment clues "don't", representing negative sentiment. For a sentence that contains two aspects (like fourth sentence *#4*), when the aspect is "dish" and it does not have explicit opinion expressions, the DualGCN and BiSyn-GAT models may suffer from the noisy interference introduced by attention mechanism and syntactic dependency tree, leading to the deviation in terms of locating aspect and its corresponding expressions. For a sentence that contains three aspects (like fifth sentence *#5*), when the aspect is "sushi" and it does not also have explicit opinion expressions, both Du-

alGCN and BiSyn-GAT models align aspect "sushi" and sentiment expression "excellent", and thus predict its sentiment polarity is positive, which is the wrong result. Because the expression "excellent" is the corresponding opinion word of aspect "rolls". Both DualGCN and BiSyn-GAT models are mislead by the sentiment expressions related to other aspects. Therefore, compared with other models, our A2SMvCL model can directly eliminate the interference of aspect-unrelated words and capture significant aspect-dependent sentiment clues more accurately. In addition, these cases also demonstrate that our A2SMvCL model can achieve better performance in the scenario with multiple aspects.

## B  More Visualization Results

Figure 7 and Figure 8 show the visualization results on REST14 and TWITTER datasets. We show the t-SNE visualization of intermediate embeddings learned by BiSyn-GAT and our A2SMvCL models. Again, the visualization results show that the embeddings within the same sentiment are more aggregated, and the boundaries between different sentiments are more distinct. This verifies that A2SMvCL can derive more definite correlation and clearer difference among different sentiments, which is more conducive to the ABSA task.