# OpenReview forum: "Aspect-to-Scope Oriented Multi-view Contrastive Learning  for Aspect-based Sentiment Analysis"
_EMNLP/2023/Conference — EMNLP 2023 Findings_

### Official Review · Reviewer_viZs · 2023-08-04

**Soundness:** 3

**Excitement:**

2: Mediocre: This paper makes marginal contributions (vs non-contemporaneous work), so I would rather not see it in the conference.

**Missing References:**

1. Binxuan Huang and Kathleen M Carley. Syntax Aware Aspect Level Sentiment Classification with Graph Attention Networks. EMNLP-2019
2. Amir Pouran Ben Veyseh, Nasim Nouri, Franck Dernoncourt, Quan Hung Tran, Dejing Dou, Thien Huu Nguyen. Improving Aspect-based Sentiment Analysis with Gated Graph Convolutional Networks and Syntax-based Regulation. EMNLP-2020: findings
3. Yuanhe Tian, Guimin Chen, Yan Song. Aspect-based Sentiment Analysis with Type-aware Graph Convolutional Networks and Layer Ensemble. NAACL-2021

**Paper Topic And Main Contributions:**

The paper proposes an approach named aspect-to-scope multi-view contrastive learning (A2SMvCL) to improve Aspect-Based Sentiment Analysis (ABSA), where syntactic and semantic knowledge are leveraged through graph-based method. The model also contrastive learning to distinguish between aspect-related and unrelated sentiment representations. Experiments conducted on six public benchmark datasets show good performance of the proposed A2SMvCL.

**Questions For The Authors:**

1. In the experience, did the authors use BERT-base or BERT-large?
2. How did the authors tune the hyper-parameters?
3. Did the author try other parsers to get the dependencies? Will the quality of the dependencies affect model performance?


**Reasons To Accept:**

1. The paper proposed an approach for ABSA
2. The proposed approach achieved good results on six English benchmark datasets.


**Reasons To Reject:**

1. The techniques used in the paper are not new. Both syntactic and semantic features are extensively employed in existing ABSA research. Additionally, graph-based approaches are also widely used for encoding these features. Furthermore, contrastive learning is not an innovative concept.
2. The approach is complicated and basically is the combination of existing techniques with minor changes.
3. Some important technique details are missing. (see questions)
4. It is not clear how the hyper-parameters are tuned. It seems the result reported in the paper comes from a single run (the authors do not report the average and standard deviation of the results). Given the test sets of the benchmark datasets are small, it is possible that the model overfits the test set.
5. Many related studies are not discussed. (examples are in missing references)

**Reproducibility:**

3: Could reproduce the results with some difficulty. The settings of parameters are underspecified or subjectively determined; the training/evaluation data are not widely available.

**Reviewer Confidence:**

4: Quite sure. I tried to check the important points carefully. It's unlikely, though conceivable, that I missed something that should affect my ratings.

---

> ### Author Rebuttal · Authors · 2023-08-28
>
> We sincerely appreciate your time in reading the paper, and our point-to-point responses to your comments are given below.
>
> Q1: The techniques used in the paper are not new. The approach is complicated and basically is the combination of existing techniques with minor changes.
>
> A1: Thanks for the comments. and we partially agree with you. The use of syntactic and semantic feature, as well as graph-based approaches, is widely used in ABSA research. Our main contribution is not to improve these existing techniques. We employ these techniques to help obtain the syntactic and semantic dependency relationship. We focus on addressing the challenge that when multiple aspect terms exist in a sentence, the opinion expressions corresponding to these aspect terms may interfere with each other, resulting in false predictions.
>
> To address this issue, we use the aspect-specific scope to separate different aspect-related context words to prevent noisy interference from irrelevant context words. Meanwhile, we propose a novel multi-view graph contrastive learning framework to accurately align each aspect and its corresponding opinion expressions so that obtains semantic- and syntactic-enhanced aspect-oriented sentiment representation.
>
> This is the first work to leverage contrastive learning to fuse different types of information within a sentence while separating different aspect-dependent sentiment representations in syntactic and semantic representation space. Moreover, the construction of positive/negative samples and contrastive learning strategy are novel. We first define positive/negative samples from a new viewpoint on whether words are in/out aspect-specific scopes. Then, we propose intra- and inter-view contrastive learning objects to facilitate the learning of the richer aspect-dependent and sentiment-aware representation, which is also novel in the social media text analysis area.
>
> We hope that this explanation clarifies the novelty of our approach.
>
> Q2: It is not clear how the hyper-parameters are tuned. It seems the result reported in the paper comes from a single run (the authors do not report the average and standard deviation of the results). Given the test sets of the benchmark datasets are small, it is possible that the model overfits the test set.
>
> A2: We appreciate the question and would like to explain the confusion about the experimental details. The results in Table 2 are the average performance over five runs by using different seeds. We have revised the results in Table 2 and supplemented the standard deviation of the results. The performance of our A2SMvCL on the **Accuracy** metric is presented below.
>  $$
> \begin{array}{ccccccc}
> \text{Model} & \text{Lap14} & \text{Rest14} & \text{Rest15} & \text{Rest16} & \text{Twitter}  & \text{MAMS} \\\\
> \hline
> \text{A2SMvCL(Ours)} & 82.12\pm0.27 & 87.86\pm0.22 & 86.74\pm0.56 & 93.42\pm0.28 & 78.49\pm0.74 & 85.10\pm0.30 \\\\
>  \end{array}
> $$
> The performance of our A2SMvCL on the  **F1-score** metric is presented below.
>  $$
> \begin{array}{ccccccc}
> \text{Model} & \text{Lap14} & \text{Rest14} & \text{Rest15} & \text{Rest16} & \text{Twitter}  & \text{MAMS} \\\\
> \hline
> \text{A2SMvCL(Ours)} & 78.82\pm0.18 & 82.41\pm0.27 & 75.05\pm2.13 & 83.80\pm1.25 & 77.18\pm0.13 & 84.65\pm0.21 \\\\
>  \end{array}
> $$
>
> Similar to all comparison methods, we follow the standard training/testing set split to conduct all experiments. Although the testing sets are small, the dataset publisher can ensure that they were randomly chosen and should represent the distribution of the entire dataset. Moreover, we tested our proposed A2SMvCL method on different datasets to comprehensively evaluate its performance.
>
> In addition, during the experimental phase, the model's training and testing processes were independent. We did not use the test set for any kind of parameter tuning, thereby ensuring the model's generalization to the test set.
>
> Q3: Many related studies are not discussed. (examples are in missing references).
>
> A3: We appreciate the suggestion and will cite and analyze these works in our revision.
>
> Q4: In the experience, did the authors use BERT-base or BERT-large?
>
> A4: Thanks for the comments. In this paper, for a fair comparison, we follow the existing ABSA methods and use the identical **BERT-base** model as the text encoder.
>
> Q5: How did the authors tune the hyper-parameters?
>
> A5: Thanks for the comments. Due to some mistakes in our writing, there is ambiguity about hyper-parameters. In fact, we only introduce two hyper-parameters $\gamma_g$ and $\tau$ to control and balance our proposed scope-assisted multi-view GCL strategy. The $\gamma_c$ and $\gamma_s$ are not hyper-parameters and are set empirically as 1 and 0.03, respectively, according to the HGCN model [1].
>     We adopt the Grid Search technique to tune the hyper-parameters. We empirically select a set of possible values for the hyper-parameters, then perform combinations on these values to find the optimal hyper-parameter combination on all datasets.
>     Grid Search technique is a common parameter tuning technique, and it can systematically explore all possible combinations to find the optimal values of hyper-parameters.
>
> Q6: Did the author try other parsers to get the dependencies? Will the quality of the dependencies affect model performance?
>
> A6: Thanks for the comments. To evaluate the impact of different parsers, we conduct an experiment based on our proposed A2SMvCL by using two common dependency parsers, Stanford Parser [2] and Biaffine Parser [3]. The results are presented below. The best results are bold. From the following table, we can find that using Stanford Parser can obtain better performance of ABSA task. Moreover, these two dependency parsers have little impact on the performance of our proposed A2SMvCL method.
>  $$
> \begin{array}{cccc}
> \text{Parser} & \text{Lap14} & \text{Rest14} & \text{Twitter} \\\\
> \hline
> \text{Stanford} & \boldsymbol{82.12} & \boldsymbol{87.86} & \boldsymbol{78.49} \\\\
> \text{Biaffine} & 81.97 & 87.23 & 78.08 \\\\
>  \end{array}
> $$
>
> [1] Lvxiaowei Xu, Xiaoxuan Pang, Jianwang Wu, Ming Cai, and Jiawei Peng. 2022. Learn from structural scope: Improving aspect-level sentiment analysis with hybrid graph convolutional networks. arXiv preprint arXiv:2204.12784.
>
> [2] Danqi Chen and Christopher Manning. A fast and accurate dependency parser using neural networks. EMNLP.
>
> [3] Timothy Dozat and Christopher D Manning. Deep biaffine attention for neural dependency parsing. ICLR.

---

### Official Review · Reviewer_PEgC · 2023-08-06

**Soundness:** 3

**Excitement:**

3: Ambivalent: It has merits (e.g., it reports state-of-the-art results, the idea is nice), but there are key weaknesses (e.g., it describes incremental work), and it can significantly benefit from another round of revision. However, I won't object to accepting it if my co-reviewers champion it.

**Paper Topic And Main Contributions:**

The paper studies aspect-based sentiment analysis (ABSA) and focused on eliminating noise introduced by the attention mechanism and dependency tree. To address the issue, the paper proposed a scope-assisted multi-view group contrastive learning framework (A2SMvCL), which contains five components: 1. aspect scope detection, 2. graph generation, 3. contrastive learning, 4, graph fusion, and 5. aspect-oriented sentiment classification.  On six benchmark datasets, the paper reports improvements against various baselines.

**Reasons To Accept:**

1. Improved performance on six benchmark datasets over various baselines. The improvements seem to be significant.
2. Thorough experiments and analysis.

**Reasons To Reject:**

1. It is hard to connect the problem to model building. There could be multiple paths to address the problem, but it is not clear why selecting graphs and contrastive learning. For example, if the noise is introduced by syntax tree and/or attention maps, why build contrastive learning on top of H rather than the input of H, i.e. eliminating noise from the early stage.

**Reproducibility:**

3: Could reproduce the results with some difficulty. The settings of parameters are underspecified or subjectively determined; the training/evaluation data are not widely available.

**Reviewer Confidence:**

3: Pretty sure, but there's a chance I missed something. Although I have a good feel for this area in general, I did not carefully check the paper's details, e.g., the math, experimental design, or novelty.

---

> ### Author Rebuttal · Authors · 2023-08-27
>
> Thanks for your encouraging words and constructive comments. We sincerely appreciate your time in evaluating our work. Our point-to-point responses to your comments are given below.
>
> Q1: It is hard to connect the problem to model building. There could be multiple paths to address the problem, but it is not clear why selecting graphs and contrastive learning.
>
> A1: Thanks for the comments. The main challenge of ABSA is how to accurately locate the aspect term and its corresponding opinion expressions while preventing interference from irrelevant context words. The graph structure is a very intuitive and effective way of modeling the relationship between word and word. Therefore, many ABSA methods use the graph to boost their performance, demonstrating the superiority and effectiveness of graphs in capturing the semantic representation of sentences. Consequently, we use the graph to help capture richer semantic information from sentences.
>
> The motivation of this paper is that when multiple aspect terms exist in a sentence, the opinion expressions corresponding to these aspect terms may interfere with each other, resulting in false predictions. Accordingly, we use the aspect-specific scope to separate different aspect-related context words to prevent noisy interference from other aspects.  However, relying solely on the aspect-specific scope is inadequate for capturing a richer aspect-dependent sentiment representation, given that each aspect-specific scope carries limited semantic and syntactic information.
> Therefore, we propose a multi-view graph contrastive learning approach to enhance aspect-dependent sentiment-aware representation by integrating syntactic dependency and semantic correlation information.
>
> Q2: For example, if the noise is introduced by syntax tree and/or attention maps, why build contrastive learning on top of H rather than the input of H, i.e. eliminating noise from the early stage.
>
> A2: Thanks for the comments. We need to leverage syntax tree and attention mechanism techniques to capture syntactic and semantic dependency relationships in a sentence, respectively. After that, we use contrastive learning to enhance the aspect-dependent sentiment representation, so as to obtain high-quality aspect-dependent sentiment representation. If we build contrastive learning on the input of ${H}$, it may limit the ability of the model to acquire high-quality aspect-dependent syntactic and semantic dependency features, thus hindering the learning of a more nuanced aspect-dependent, sentiment-aware representation.
> We hope that this explanation clarifies the rationale behind our approach. We will include these details in the revised manuscript for better understanding.

---

### Official Review · Reviewer_etwT · 2023-08-08

**Soundness:** 3

**Excitement:**

3: Ambivalent: It has merits (e.g., it reports state-of-the-art results, the idea is nice), but there are key weaknesses (e.g., it describes incremental work), and it can significantly benefit from another round of revision. However, I won't object to accepting it if my co-reviewers champion it.

**Paper Topic And Main Contributions:**

The paper discusses the construction of syntax-based and semantic-based graphs for aspect-dependent sentiment analysis. The paper proposes the use of GCN encoders to capture syntactic and semantic representations. It also introduces a scope-assisted multi-view graph contrastive learning approach to align aspect and opinion words while separating in-scope and out-scope words. The paper further describes intra-view and inter-view contrastive learning techniques. Extensive experiments show effectiveness and robustness.

**Questions For The Authors:**

I think the improvement over KumaGCN is tiny on several datasets. Can you give some further analysis?

**Reasons To Accept:**

1. The paper provides a comprehensive approach for aspect-based sentiment analysis, considering both syntax and semantics.
2. The paper introduces a scope-assisted multi-view graph contrastive learning approach, which aligns aspect and opinion words while separating in-scope and out-scope words.
3. The paper presents intra-view and inter-view contrastive learning techniques to enhance the performance of sentiment analysis.
4. The experimental results demonstrate that the proposed framework outperforms several comparison baselines on multiple datasets, indicating its effectiveness in the ABSA task.

**Reasons To Reject:**

It says in the abstract "Most existing ABSA methods... suffers from noisy interference introduced by the attention mechanism and dependency tree when multiple aspects exist in a sentence". I would like to challenge this because the dependency tree is not used in many ABSA methods, so the problem proposed in this paper may not be an important issue for a lot of ABSA methods.


**Reproducibility:**

4: Could mostly reproduce the results, but there may be some variation because of sample variance or minor variations in their interpretation of the protocol or method.

**Reviewer Confidence:**

4: Quite sure. I tried to check the important points carefully. It's unlikely, though conceivable, that I missed something that should affect my ratings.

---

> ### Author Rebuttal · Authors · 2023-08-27
>
> We sincerely appreciate your time in reading the paper, and our point-to-point responses to your comments are given below.
>
> Q1: I would like to challenge this because the dependency tree is not used in many ABSA methods, so the problem proposed in this paper may not be an important issue for a lot of ABSA methods.
>
> A1: Sorry for the confusion. A writing error leads to a misunderstanding. Existing ABSA methods use either the dependency tree technique or the attention mechanism technique to improve the performance of ABSA. Either of these two techniques may bring up noisy interference problems.
> Indeed, the dependency tree is not used in all ABSA methods. But, the attention mechanism technique is very common in ABSA. Most ABSA methods use attention mechanism to capture the aspect-related semantic
>  information. Therefore, the proposed problem in this paper is an important issue in ABSA methods.
>
> Q2: I think the improvement over KumaGCN is tiny on several datasets. Can you give some further analysis?
>
> A2: Thanks for the comments. KumaGCN used the dependency tree to obtain syntactic dependencies representation and the attention mechanism to derive the latent graph. Our proposed A2SMvCL approach obtains a tiny improvement over KumaGCN on \textsc{Lap14} and \textsc{Twitter} datasets. The major reason is that these two datasets contain fewer multi-aspect sentences: \textsc{Lap14} dataset has 36\% multi-aspect sentences, and \textsc{Twitter} dataset does not contain multi-aspect sentences. Therefore, the latent graph proposed by KumaGCN can accurately locate the aspect term and its corresponding opinion expression, resulting in comparable performance on single-aspect datasets.
>     Moreover, our A2SMvCL significantly improved compared to the KumaGCN method on other datasets that contain more multi-aspect sentences. This demonstrates that when multiple aspects exist in a sentence, KumaGCN may suffer from noisy interference caused by multiple aspects.
>     We hope that this explanation clarifies the confusion behind the KumaGCN method. We will include these details in the revised manuscript for better understanding.

---

### Official Review · Reviewer_WYcF · 2023-08-12

**Soundness:** 3

**Excitement:**

3: Ambivalent: It has merits (e.g., it reports state-of-the-art results, the idea is nice), but there are key weaknesses (e.g., it describes incremental work), and it can significantly benefit from another round of revision. However, I won't object to accepting it if my co-reviewers champion it.

**Paper Topic And Main Contributions:**

This paper propose a scope-assisted multi-view graph contrastive learning framework called A2SMvCL for ABSA task. Experiments on six benchmark datasets show that this approach outperforms state-of-the art methods.

**Questions For The Authors:**

A. Figure 1 (a) shows the attention scores for the anchor aspect “people” to its context. How does these attention scores get?

B. As mentioned in Reasons To Reject, could you give some explanation about the extra annotation problem?

**Reasons To Accept:**

The paper is well structured and clear to follow.

The scope-assisted multi-view (intra- and inter-view) graph contrastive learning effectively extracts high-quality features and this approach can be extended to other tasks.

The proposed model achieves the state-of-the-art on several datasets and verifies the effectiveness.

**Reasons To Reject:**

The scope labels need extra annotation. This means that the method proposed in this paper requires more expensive labour costs for practical application, and makes the comparison between this method and baseline methods less fair.

Of course, if scope labels can be annotated fully automatically, the above problem does not exist. However, in the paper presenting the dataset with scope labels[1], it says "According to our statistics, 26.4% of the samples require manual adjustments." I am concerned that the effectiveness of the method in this paper may be compromised by low-quality automatically annotated scope label.

[1] Lvxiaowei Xu, Xiaoxuan Pang, Jianwang Wu, Ming Cai, and Jiawei Peng. 2022. Learn from structural scope: Improving aspect-level sentiment analysis with hybrid graph convolutional networks. arXiv preprint arXiv:2204.12784.

**Reproducibility:**

3: Could reproduce the results with some difficulty. The settings of parameters are underspecified or subjectively determined; the training/evaluation data are not widely available.

**Reviewer Confidence:**

4: Quite sure. I tried to check the important points carefully. It's unlikely, though conceivable, that I missed something that should affect my ratings.

---

> ### Author Rebuttal · Authors · 2023-08-28
>
> Thanks for your encouraging words and constructive comments. We sincerely appreciate your time in evaluating our work. Our point-to-point responses to your comments are given below.
>
> Q1: The scope labels need extra annotation. This means that the method proposed in this paper requires more expensive labour costs for practical application....
>
> A1: Thanks for the comments. The scope labels can be acquired automatically by using the annotation tool [1]. In this paper, the used scope-assisted datasets are provided by Lvxiaowei Xu et al. [1].
>
> [1] Lvxiaowei Xu, Xiaoxuan Pang, Jianwang Wu, Ming Cai, and Jiawei Peng. 2022. Learn from structural scope: Improving aspect-level sentiment analysis with hybrid graph convolutional networks. arXiv preprint arXiv:2204.12784.
>
> Q2: Of course, if scope labels can be annotated fully automatically, the above problem does not exist. However, in the paper presenting the dataset with scope labels[1], it says "According to our statistics, 26.4\% of the samples require manual adjustments." I am concerned that the effectiveness of the method in this paper may be compromised by low-quality automatically annotated scope label.
>
> A2: We appreciate your concern regarding the annotation of scope labels and the potential impact of low-quality automatic labels on the effectiveness of our method. We partially agree with you and the low-quality annotated data may affect the effectiveness of the AI model. Thanks to their efforts [1], [1] published the high-quality and manually adjusted annotated scope datasets, such as \textsc{Lap14}, \textsc{Rest14}, \textsc{Rest15}, \textsc{Rest16}, and \textsc{Mams} datasets. We use these high-quality datasets to develop a robust model that can handle some annotation errors, such as using \textit{graph structure} to handle grammatical connection interruption errors and using the \textit{Adaptive Multi-graph Fusion} strategy to handle the representation error caused by misannotation scope label. For short-sighted errors, it is hard to handle since scope fails to capture implicit transitive descriptions related to the target term. This may affect the model's performance.
>
> To further illustrate the robustness of our proposed method, we conduct a cross-domain experiment. we first train a source model on the \textsc{Mams} dataset and test its performance on the other four datasets shown in Figure 3. The results illustrate that our proposed approach outperforms three comparison methods across all four datasets, illustrating the robustness of our model A2SMvCL in learning the aspect-dependent sentiment representation. This implies that the scope-assisted multi-view graph contrastive learning framework has strong robustness for grasping aspect-opinion interactions.
>
> Q3: Figure 1 (a) shows the attention scores for the anchor aspect “people” to its context. How does these attention scores get?
>
> A3: Thanks for the comments. We construct a simple model, a pre-trained BERT-base followed by a self-attention layer, to calculate the attention scores of aspect terms.
>
> Q4: As mentioned in Reasons To Reject, could you give some explanation about the extra annotation problem?
>
> A4: We give the detailed explanation in **A2**.

---

### Meta-Review · Area_Chair_t3nQ · 2023-09-23

**Recommendation:** 3

**Metareview:**

ABSA is such a well-explored problem. The authors focus on eliminating the "noisy interference" issues in ABSA introduced by usual methods (dependency tree, attention) by proposing a scope-assisted multi-view graph contrastive learning framework. The paper is comprehensive, structured and easy to follow. The reviewers agree that the authors achieved state-of-the-art results on multiple datasets. In terms of the novelty of the task, as I mentioned, it is a well-explored one; however, their methods supposedly overcome the interference problem. The responses made by the authors to the reviewer's questions seem justified. The paper might find an audience.

---

### Decision · Program_Chairs · 2023-10-07

**Decision:**

Accept-Findings

**Comment:**

ABSA is such a well-explored problem. The authors focus on eliminating the "noisy interference" issues in ABSA introduced by usual methods (dependency tree, attention) by proposing a scope-assisted multi-view graph contrastive learning framework. The paper is comprehensive, structured and easy to follow. The reviewers agree that the authors achieved state-of-the-art results on multiple datasets. In terms of the novelty of the task, as I mentioned, it is a well-explored one; however, their methods supposedly overcome the interference problem. The responses made by the authors to the reviewer's questions seem justified. The paper might find an audience.